# Targeting NADPH Oxidase as an Approach for Diabetic Bladder Dysfunction

**DOI:** 10.3390/antiox13101155

**Published:** 2024-09-24

**Authors:** Tammyris Helena Rebecchi Silveira, Fábio Henrique Silva, Warren G. Hill, Edson Antunes, Mariana G. de Oliveira

**Affiliations:** 1Laboratory of Pharmacology, São Francisco University (USF), Bragança Paulista, Sao Paulo 12916-900, Brazil; tammyris.silveira@mail.usf.edu.br (T.H.R.S.); fabio.hsilva@usf.edu.br (F.H.S.); 2Laboratory of Voiding Dysfunction, Department of Medicine, Beth Israel Deaconess Medical Center, Boston, MA 02215, USA; whill@bidmc.harvard.edu; 3Department of Translational Medicine, Pharmacology Area, Faculty of Medical Sciences, University of Campinas (UNICAMP), Campinas, Sao Paulo 13083-970, Brazil; eantunes@unicamp.br

**Keywords:** NADPH oxidase, redox signaling, urinary bladder, incontinence, micturition, urology

## Abstract

Diabetic bladder dysfunction (DBD) is the most prevalent complication of diabetes mellitus (DM), affecting >50% of all patients. Currently, no specific treatment is available for this condition. In the early stages of DBD, patients typically complain of frequent urination and often have difficulty sensing when their bladders are full. Over time, bladder function deteriorates to a decompensated state in which incontinence develops. Based on studies of diabetic changes in the eye, kidney, heart, and nerves, it is now recognized that DM causes tissue damage by altering redox signaling in target organs. NADPH oxidase (NOX), whose sole function is the production of reactive oxygen species (ROS), plays a pivotal role in other well-known and bothersome diabetic complications. However, there is a substantial gap in understanding how NOX controls bladder function in health and the impact of NOX on DBD. The current review provides a thorough overview of the various NOX isoforms and their roles in bladder function and discusses the importance of further investigating the role of NOXs as a key contributor to DBD pathogenesis, either as a trigger and/or an effector and potentially as a target.

## 1. Introduction

Known as diabetic bladder dysfunction (DBD), diabetic uropathy, or diabetic cystopathy, this complication of diabetes affects up to 50% of men and women with chronic and poorly controlled diabetes mellitus (DM) [1,2,3]. This places DBD as the most common diabetic complication experienced by patients, and yet, it has received less attention than some of the better-known and most bothersome complications, such as retinopathy, neuropathy, and nephropathy. This disparity might stem from DBD being perceived as less severe or life-threatening than other complications. However, it is widely acknowledged that it significantly affects the quality of life of affected individuals [2].

Considered an umbrella term, DBD encompasses a broad range of signs and symptoms associated with urinary function in DM patients. Classically described as a triad of reduced bladder sensation, increased bladder capacity, and impaired detrusor smooth muscle contractility [4], the modern clinical DBD presentation is heterogeneous and includes a variety of urodynamic findings such as detrusor overactivity, detrusor underactivity, and urethral dysfunction [2,5,6]. Despite the prevalence and impact of DBD, no comprehensive classification scheme for phenotype DBD has yet been proposed, indicating a significant gap in our understanding and approach to this disorder.

From a treatment perspective, DBD is managed using conventional approaches for lower urinary tract dysfunction, such as antimuscarinics, α1-adrenoceptor antagonists, and β3-adrenoceptor agonists, with the assumption of comparable efficacy in both diabetic and non-diabetic patients [7]. When pharmacotherapy fails to improve voiding function, bladder catheterization is necessary to achieve adequate bladder emptying [1]. Unfortunately, none of these treatments target the primary causes of DBD, as its pathophysiology is poorly understood. The pathophysiology of DBD is multifactorial (reviewed in [8]) and includes three major drivers: detrusor smooth muscle dysfunction, urothelial dysfunction, and neuronal dysfunction. Systemic and circulating factors such as hyperglycemia, inflammation, advanced glycation end-products, and reactive oxygen species (ROS) are additional contributors to tissue damage. Exploration of these underlying mechanisms is essential for developing more effective, targeted therapies that address the specific pathophysiological processes of DBD.

## 2. Role of Oxidative Stress in DBD

Oxidative stress is an imbalance between oxidants and antioxidants, with a deficit in antioxidants leading to the disruption of redox signaling and molecular damage [9]. Most cells possess intrinsic defense mechanisms involving various enzymes, such as superoxide dismutase (SOD), glutathione peroxidase (GPX), and catalase (CAT), as well as several reducing cofactors including NADH, NADPH, glutathione, and tetrahydrofolate, whose levels help to manage redox potential [10,11]. However, reductions in antioxidant status have been observed in patients with type 2 DM compared to non-diabetic individuals [12]. Metabolic disturbances in DM cause excessive production of reactive oxygen and nitrogen species (ROS/RNS), leading to oxidative stress and cellular damage [13]. (Figure 1 depicts the main oxidative pathways involved in diabetic complications).

As examples, higher levels of oxidative markers, such as malondialdehyde (MDA) [14,15], advanced oxidation products of proteins [15], protein carbonyl [16], 8-OHdG [14,17], and indoxyl sulfate [18], are observed in patients with DM compared to healthy subjects. Moreover, ROS has been proposed to play a causal role in some of the most clinically significant end-organ complications of DM, such as chronic kidney disease, retinopathy, heart attack, and stroke [19,20]. For instance, in rodent models of diabetic kidney disease, oxidative stress has been associated with increased levels of nuclear factor kappa B (NF-kB), tumor necrosis factor-alpha (TNF-α), and transforming growth factor beta-1 (TGF-β1) [21,22,23,24], while antioxidant systems are impaired [25,26,27]. This imbalance ultimately leads to renal glomerular sclerosis and interstitial fibrosis [27,28]. In isolated glomerular endothelial cells exposed to high glucose and serum from diabetic mice, there was increased mitochondria-derived ROS, secretion of pro-apoptotic factors, and reduced autophagy [29]. Clinical studies with DM patients further demonstrate that renal oxidative stress contributes to the progressive loss of the glomerular filtration barrier and fibrosis in the tubulointerstitial regions of the kidneys [30,31]. However, there is a lack of research regarding the impact of DM on the human urinary bladder.

Studies addressing oxidative stress in urinary bladder function in health and disease are extremely limited, especially with human samples. One informative study showed that exogenous H_2_O_2_ (100 µM) increased contractions in human bladder strips [32]. However, much of our understanding of the role of ROS in bladder function is derived from preclinical evidence [33,34]. Direct induction of oxidative stress in the rat bladder through intravesical administration of a low dose of H_2_O_2_ (0.003%) led to detrusor hyperactivity, while conversely, a high dose of H_2_O_2_ (3%) impaired detrusor contractility [35].

A number of studies in different models have highlighted the role of ROS in DBD. In rats, the injection of streptozotocin (STZ), which is selectively toxic to pancreatic beta cells, induces DBD, along with increased thiobarbituric acid reactive substance (TBARS) levels, higher inducible nitric oxide synthase (iNOS) protein expression, and an increased number of apoptotic cells in the diabetic bladder [36]. Statistically significant increases in all three isoforms of nitric oxide synthases and increased protein nitrotyrosylation were observed as early as 3 days [37] and were maintained for at least 5 weeks after the induction of diabetes [38]. Comparative gene expression in the rat bladder of non-diabetic and STZ-induced DM (2 months) showed significant changes in genes that are involved in the modulation of oxidative stress, activation of protein degradation pathways, and apoptosis [39]. The same study also showed increased levels of lipid peroxidation and nitrotyrosylated proteins along with reduced glutathione S-transferase activity in diabetic animals. In mice, increased bladder nitrotyrosine levels were found at 9 weeks [40], 20 weeks [41], and 44 weeks [42] after STZ-DM induction. The smooth muscle-specific knockout of superoxide dismutase (SOD), one of the most critical ROS-scavenging enzymes, led to deteriorated bladder function and enhanced apoptotic signaling in the bladders of STZ-induced diabetic mice [40]. Proteomic analysis of detrusor and urothelium samples from TallyHo diabetic mice (a type 2 DM model) revealed significant dysregulation of several proteins involved in complexes I, III, and IV of the mitochondrial electron transport chain [43]. Additionally, the protein expression of key players in the Nuclear factor erythroid 2-related factor 2 (Nrf2), a prime regulator of cellular responses against oxidative stress [44], was significantly downregulated only in the urothelium but not in the detrusor [43]. Notably, compounds that activate the Nrf2 pathway, such as grape seed extract [45], sulforaphane [46], and the near-infrared dye IR-61 [47], have shown protective effects against STZ-induced DBD in rats.

Taken together, these findings demonstrate that oxidative stress induces pathophysiological changes in bladder tissues. However, there is still insufficient evidence to conclusively establish a relationship between oxidative stress and DBD initiation, as it remains unclear whether oxidative stress is a causative factor for DBD or a secondary manifestation of other underlying pathophysiological mechanisms. Additionally, in most studies, ROS is employed as a generic term, although its biological impact critically depends on the specific molecule(s) involved (e.g., NO, O_2_^−^, H_2_O_2_, and ONOO^−^). Currently, the identification of the specific deleterious ROS/RNS involved and their sources forms a reasonable basis for mechanistic studies and targeted therapies that leave other essential physiological pathways intact. ROS are now recognized as modulators of numerous cell-signaling pathways via redox-based mechanisms, which is far from the traditional viewpoint of their role as non-specific damaging molecules [48]. This implies that non-specific ROS elimination may counteract any beneficial physiological effects that could be achieved by more selective targeting.

## 3. Role of NOXs in Micturition

The major sources of ROS include the mitochondrial electron transport chain, nicotinamide adenine dinucleotide phosphate (NADPH) oxidase (NOX), xanthine oxidase (XO), cytochrome P450 enzymes, and uncoupled endothelial nitric oxide synthase (eNOS) [49] (Figure 1). Unlike other ROS sources, NOX enzymes are the only family known to produce ROS as their primary and sole function [50]. All other ROS sources serve distinct primary functions, and ROS production typically occurs after damage or uncoupling [51]. This unique catalytic activity of NOXs is the rationale for a review highlighting their role in DBD. Additionally, crosstalk between different sources of ROS is thought to be critical to a regenerative cycle of ROS-induced ROS release, whereby newly formed NOX-derived ROS may trigger and enhance further ROS formation (ROS-induced ROS-release) [50].

NADPH oxidases are transmembrane enzyme complexes that catalyze the reduction of O_2_ to ROS, specifically O_2_^−^ or H_2_O_2_, coupled with the oxidation of NADPH. Seven enzyme isoforms in the NOX family (NOX1, NOX2 (aka gp91phox), NOX3, NOX4, NOX5, DUOX1, and DUOX2) have been identified and found to be expressed in specific cell types and tissues, underlying their specific roles in health and disease [50]. In humans, all seven enzymes are expressed, while mice and rats express NOX1–4 and DUOX isoforms but lack NOX5 [50,52]. The expression levels of NOXs are frequently regulated by growth factors and cytokines as well as by other stimuli such as hypoxia, low pH, and bacterial products [53]. Their structures vary between isoforms but, in general, consist of a conserved six-helix transmembrane domain and an N-terminal dehydrogenase domain located intracellularly, which provides binding sites for the NADPH substrate and the co-factor FAD (flavin adenine dinucleotide). The oxidase complex is also regulated by cytosolic regulatory proteins, including p47, p67, p40, and the small GTPases Rac1 or Rac2, and, in the case of NOX5, DUOX1, and DUOX2, the regulation also involves a calcium-binding domain [52]. For the interested reader, the subcellular localization and activation mechanisms of NOXs have been reviewed elsewhere [53], as well as their isoform-specific roles across different tissues and conditions, influencing various pathophysiological responses [51]. NOX-derived ROS contribute to physiological functions and processes, such as the oxidative burst of the innate immune response, cell proliferation, differentiation, migration, ion channel conductance, vasodilation, hearing, insulin secretion, and insulin sensitivity (for review, see [9]). Nevertheless, the role of NOX-derived ROS, even in the healthy bladder, remains poorly understood. This gap highlights the need for targeted research to elucidate the specific contributions of NOXs in the urinary system.

In humans, only one study has been conducted on NOXs, and this showed that NOX4 is not expressed in the normal urothelium but is increased in samples from patients with urothelial carcinoma [54]. In mice, studies have shown that both NOX2 and NOX4 mRNA are expressed in the bladder [55], urethra [56], and prostate [57]. These studies showed that NOX2 expression was dramatically increased in these organs during inflammation [55] and in the setting of obesity [56,57], with increased levels of O_2_^−^ correlated with dysfunction. However, the samples studied consisted of whole-bladder homogenates; thus, the localization (s) of NOX mRNA remained undetermined. Donkó et al. [58] showed that mouse bladder urothelial cells produce H_2_O_2_ in a calcium-dependent manner and, further, that the enzymatic source of H_2_O_2_ was urothelial DUOX1, which they confirmed using a gene-deficient mouse model. Furthermore, the absence of DUOX1 in the urothelium led to altered urinary bladder function with increased voiding frequency in relation to wild-type controls, suggesting that DUOX1-derived H_2_O_2_ plays a paracrine role in bladder detrusor function [58]. Thus, studies investigating differential NOX expression in the bladder wall are required to understand the expression and role of each NOX isoform in health and disease. For instance, in Akita transgenic mice expressing NOX5, renal fibrosis was primarily mediated by vascular smooth muscle cell- and mesangial cell-specific NOX5 expression [59]. Recently, NOX2 (but not NOX1/4) was identified as the main source of ROS overproduction in mice with cyclophosphamide-induced cystitis [55]. Further studies should focus on the role of NOX in the central regulation of micturition. NOX2 was recently localized in the periaqueductal gray and Barrington’s nucleus (pontine micturition center, PMC), which are important structures in central nervous bladder control, suggesting that NOX2-derived O_2_^−^ may be involved in the micturition reflex [60]. Figure 2 summarizes the NOX isoforms that have been identified in the lower urinary tract.

## 4. NOX Isoforms as Primary Sources of ROS during DBD

Little is known about the mechanisms relating to NOX, ROS production, and DBD. One study showed elevated Rac1 and NOX4 protein expression in rat bladders after 6 weeks of STZ-induced DM [61], but the authors did not further elucidate the contribution of NOX4 to this dysfunction. The absence of validated isoform-specific antibodies and inhibitors significantly hinders investigations into the direct involvement of NOXs in the development and progression of various diseases. Nevertheless, it is particularly noteworthy that there exists a scarcity of studies focused on bladder dysfunction.

Initially, DBD manifests as a compensated bladder state, which results in detrusor muscle hypertrophy and hypercontractility. This is followed by a decompensated state with a hypocontractile detrusor muscle. Several hyperglycemia-associated mechanisms have been identified that contribute to the development of DM complications, including ROS, excessive polyol flux, higher Advanced Glycation End (AGE) product formation, and overactivation of protein kinase C (PKC) pathways [34]. PKC is a regulatory enzyme that plays a prominent role in the signal transduction of several vascular functions in healthy individuals [62] and those with DM [63]. Under physiological conditions, detrusor muscarinic M3 receptors can couple to a range of signaling pathways, including phospholipase C (PLC), with the subsequent formation of inositol triphosphate (IP3) and diacylglycerol (DAG) to release calcium from intracellular stores and activate PKC, which in turn is coupled to L-type calcium channels [64]. PKC activation by phorbol 12, 13-dibutyrate (PDBu) increases nerve-mediated contractions in vitro and micturition contractions in vivo [65]. Enhanced PKC expression in bladder tissues has been observed in HFD-fed obese mice and DM2 mice [66]. In ob/ob mice, increased bladder contraction in response to the muscarinic agonist carbachol was shown to be dependent on the activation of the Rho-dependent PKC pathway [67]. PKC activation is mediated, at least in part, by increased ROS production [68]. The activity and expression of NOX and the phosphorylation of p47phox, a key cytosolic subunit of NOX2, are increased upon incubation with recombinant PKC isoforms or the PCK activator PMA and attenuated in PKCβ knockout mice [69,70,71], suggesting an interplay between NOX and PKC activity. In a model of STZ-induced DM, the deletion of NOX4 protected against renal injury via PKC-dependent mechanisms [72]. Taken together, these results indicate that hyperglycemia-induced activation of the PKC pathway is an additional mechanism underlying NOX-derived ROS and diabetic complications.

The most frequently assessed parameter of DBD in animal models of diabetes is bladder enlargement, which is consistently observed in type 1 and less consistently in type 2 DM models and often involves fibrosis and inflammation [73]. NOX1, NOX2, and NOX4 are involved in the initiation of liver fibrosis [74], while NOX4 mediates idiopathic pulmonary fibrosis [75]. Silencing of NOX4 results in the downregulation of pro-inflammatory and pro-fibrotic markers in diabetic nephropathy [76]. NOX activities are also critical for NLRP3 inflammasome activation [77,78], which is itself a key player in DBD [79]. To the best of our knowledge, no study has yet evaluated the role of NOX in DBD-induced fibrosis.

Very little is known about the impact of DM on the bladder urothelium. More than a barrier, the urothelium is a specialized epithelium that surveils its mechanochemical environment and communicates changes to underlying tissues, including afferent nerve fibers and smooth muscle (for review, see [80]). In rats observed at 16 weeks [81] and 3, 9, and 20 weeks [82], STZ-induced DM alters urothelial morphology, which appears rugged and disorganized. This compromised barrier function and alterations in urothelial mechanosensitivity could contribute to the altered bladder sensation in DBD [1]. Yet, no studies have focused on the role of NOXs in maintaining urothelial integrity during DBD, which remains an important area for future investigations, offering the prospect of identifying novel therapeutic targets. Recently, in a mouse model of cyclophosphamide-induced cystitis, NOX2 inhibition by GSK2795039 restored urothelial barrier integrity after injury [55].

## 5. Current Status of NOX Selective Inhibitors

Given the role of NOX in the regulation of various disease processes, blocking the undesirable actions of these enzymes is a potential therapeutic strategy for treating a number of pathological disorders. Over the years, a number of small molecules have been reported as NOX inhibitors, and these have allowed in vitro and in vivo investigations of the role of NOX-derived ROS, for instance, in cardiovascular and neurodegenerative diseases, inflammation, and cancer [83]; however, most were not specific due to off-target effects. Examples of these molecules include diphenyleneiodonium (DPI), which inhibits many other enzymes besides NOX, and apocynin, which has non-specific ROS scavenger properties [84]. In preclinical studies, only a limited number of compounds have been identified as specific NOX inhibitors. These include the NOX2 selective inhibitor GSK2795039 [85], NOX4 selective inhibitor GLX7013114 [86], APX-115, which inhibits NOX1, NOX2, and NOX4 [87], and NCATS-SM7270, a selective NOX2 inhibitor [88]. Ongoing clinical trials of NOX inhibitors include tests with GKT137839, also known as setanaxib, a preferential direct inhibitor of NOX1 and NOX4, in patients with primary biliary cholangitis and idiopathic pulmonary fibrosis (NCT05014672 and NTC03865927, respectively). However, progress in this field is challenged by the non-selective ROS-scavenging effects of most NOX inhibitors. In silico approaches, along with in vitro experiments, may accelerate the development of specific and fully validated inhibitors of NOX [89]. Additionally, significant progress has been made in the development of isoform-selective peptide inhibitors [90] and miRNAs [91].

## 6. Concluding Remarks and Future Directions

The expression of NOX isoforms and the crucial role of NOX-mediated ROS in various signaling pathways during health and disease are well established. In DM, NOXs significantly contribute to complications such as retinopathy, neuropathy, and nephropathy. However, comparatively less attention has been directed towards understanding the role of NOXs in DBD, the most common complication of DM. Previous studies have shown that NOXs are expressed in the lower urinary tract; however, their specific involvement in the development and progression of DBD remains unclear. Further research is required to elucidate the mechanisms and implications of NOX dysregulation on bladder function. Additionally, it is critical to clarify whether NOXs serve as key contributors to DBD pathogenesis, act as initiators of dysfunction or mediators of progression, and possibly contribute to the temporal changes observed. Continued research is essential to identifying the specific NOX isoforms involved in different stages of DBD and understanding their distinct roles. Consideration of NOX isoforms as primary targets in DBD might potentially drive therapeutic advances.

## Figures and Tables

**Figure 1 antioxidants-13-01155-f001:**
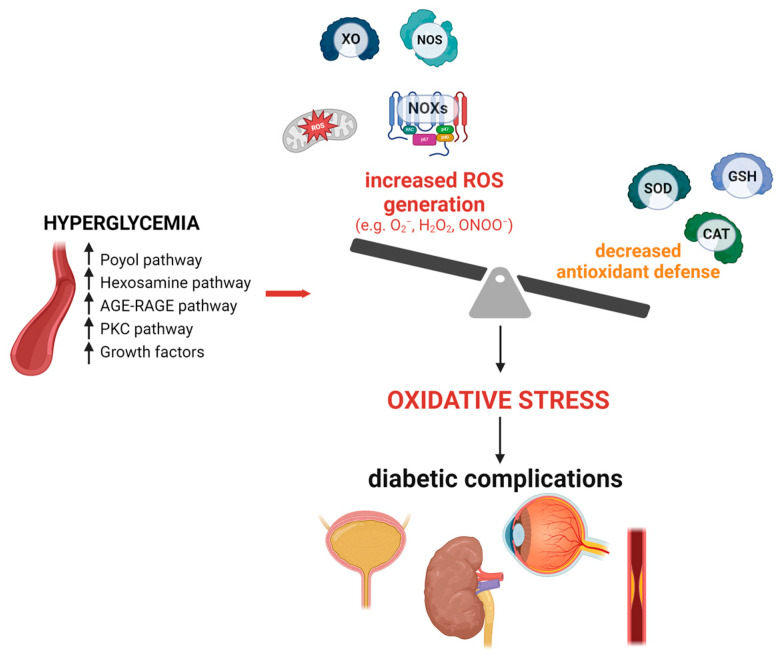
The main sources of reactive oxygen species (ROS) in diabetic complications. AGE, advanced glycation end-products; CAT, catalase; GSH, glutathione peroxidase; H_2_O_2_, hydrogen peroxide; NOX, NADPH oxidase; NOS, nitric oxide synthase; O_2_^−^, superoxide; ONOO^−^, peroxynitrite; PKC, protein kinase C; RAGE, advanced glycation end-products receptor; ROS, reactive oxygen species; SOD, superoxide dismutase; XO, xanthine oxidase. Upper arrows indicate increase.

**Figure 2 antioxidants-13-01155-f002:**
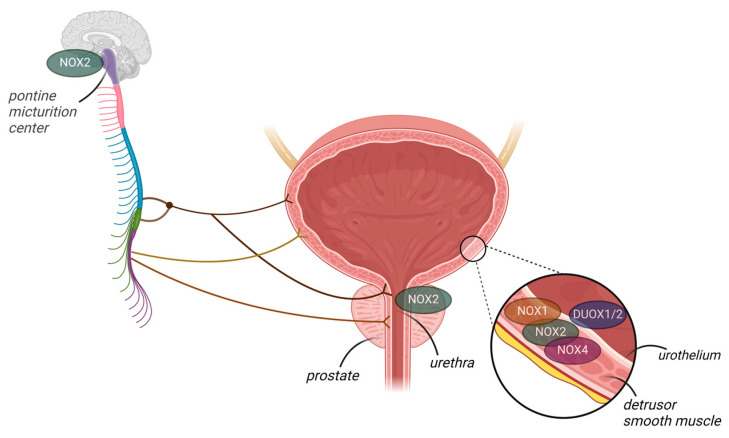
NADPH oxidase (NOX) isoforms identified in the lower urinary tract organs, though their functions remain elusive. NOX2 is expressed in the mouse pontine micturition center (PMC), bladder wall, urethra, and prostate. Additionally, NOX1 and NOX4 are expressed in the bladder wall. DUOX1/2 is present in the mouse urothelium.

## Data Availability

Not applicable.

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
