# Peer review of "Targeting NADPH Oxidase as an Approach for Diabetic Bladder Dysfunction"

_antioxidants, 2024, doi:10.3390/antiox13101155_

Round 1
Reviewer 1 Report
The paper entitled “Targeting NADPH Oxidase as an Approach for Diabetic Bladder Dysfunction” by Rebecchi Silveira and collaborators provides a comprehensive review of the knowledge of the oxidant generating NADPH oxidase in bladder diabetic complications.
Although the studies of NOX in DBD are scarce, the review provides a description of the role of oxidative mechanism in DBD, localization of NOX isoforms in the urinary system and attempts to connect molecular mechanisms known to be involved in DBD with the above.
The paper is in the scope of the journal. It is well written and pleasant to read. It is ready for publication in Antiioxidants following minor modifications.
Lane 19: “we now understand” is too strong. Prefer a more balanced verb
Lane 103: give the names of the Nrf2 activators tested in these studies
Lane 114: cell signaling
Lane 213: in fact, NOX2 was convincingly shown to repress of NLRP3 inflammasome activation (PMID: 37805023; PMID: 31954113)
Lane 238: mention NCATS-SM7270 as optimized NOX2 inhibitor (PMID: 36709665)
Lane 246: remove enzymes. It is tautologic. A Nadph Oxidase (NOX) is always an oxidase.
NA
Author Response
Firstly, we would like to thank the Reviewers for their time, consideration, critical comments, and helpful suggestions toward strengthening this review. We agree with most of their comments and have modified the text according to their main suggestions. We also thank the Editor for allowing us to submit a revised copy of the review.
Reply to Reviewer #1 comments:
Lane 19: “we now understand” is too strong. Prefer a more balanced verb.
We change the text to “it is now recognized”.
Lane 103: give the names of the Nrf2 activators tested in these studies
We appreciate the Reviewer comment and change the text to “Notably, compounds that activate the Nrf2 pathway, such as grape seed extract [32], sulforaphane [33], and the near-infrared dye IR-61 [34], have shown protective effects against STZ-induced DBD in rats”.
Lane 114: cell signaling
Corrected.
Lane 213: in fact, NOX2 was convincingly shown to repress of NLRP3 inflammasome activation (PMID: 37805023; PMID: 31954113)
We thank the Reviewer highlighting these significant studies. Indeed, NLRP3 priming and activation is inhibited by NOX2 in isolated phagocytes and THP-1 cells. However, we understand that the role of NADPH oxidase (NOX) in NLRP3 inflammasome activation is complex and still controversial. For instance, previous studies have shown that NOX4 is required for activation of the NLRP3 inflammasome [PMID: 30354670; 33584299; 26728324]. In chronic kidney disease, NOX-derived ROS can lead to kidney damage due to activation of NLRP3 inflammasome [PMID: 28193546]. In nonalcoholic fatty liver disease in mice, lactate-producing bacteria in the gut can activate NOX2 which results in NLRP3 inflammasome activation and exacerbates disease [PMID: 31217465]. In NOX2 knockout mice, the expression of NLRP3 inflammasome components (NLRP3, ASC, caspase-1, and IL-1β) was attenuated in a cerebral cortex injury [PMID: 28785377]. Therefore, in this manuscript, we have chosen to maintain the general reference to NOX as a regulator of NLRP3 without specifying the isoform or effect. We also have included a recent review on this subject:
-
- Begum R, Thota S, Abdulkadir A, Kaur G, Bagam P, Batra S. NADPH oxidase family proteins: signaling dynamics to disease management. Cell Mol Immunol. 2022 Jun;19(6):660-686. doi: 10.1038/s41423-022-00858-1.
Lane 238: mention NCATS-SM7270 as optimized NOX2 inhibitor (PMID: 36709665)
We thank the Reviewer for bringing this study to our attention. We have included it in line 275 and quoted the reference accordingly:
- Mason H, Rai G, Kozyr A, De Jonge N, Gliniewicz E, Berg LJ, Wald G, Dorrier C, Henderson MJ, Zakharov A, Dyson T, Audley J, Pettinato AM, Padilha EC, Shah P, Xu X, Leto TL, Simeonov A, Zarember KA, McGavern DB, Gallin JI. Development of an improved and specific inhibitor of NADPH oxidase 2 to treat traumatic brain injury. Redox Biol. 2023 Apr;60:102611. doi: 10.1016/j.redox.2023.102611.
Lane 246: remove enzymes. It is tautologic. A Nadph Oxidase (NOX) is always an oxidase.
Corrected.
Reviewer 2 Report
The review article entitled “Targeting NADPH Oxidase as an Approach for Diabetic Blader Dysfunction” describes the recent progress in the research area of diabetic bladder dysfunction with a focus on its relationship with ROS and NOX. Briefly, the information provided is too specific and too rough to understand an overview of the role and significance of ROS and NOXs in diabetic blader dysfunction for the reader unfamiliar with this research area. In the present form this manuscript is not informative as a review article. The English should be rechecked carefully.
1. As mentioned in the first sentence of the 2nd section (2. Role of oxidative stress in DBD), generation of ROS/RNS depends on the balance of oxidants and antioxidants. In the present form information about antioxidants including regulation of NADP+/H level is missing. As a review, providing more well known general information will be helpful for the readers, such as comparisons of the results of other well characterized tissue/organ with those of bladder under DM conditions. Representation of the relationships of the factors inducing oxidative stress and phenomena with figures and/or tables will help the understandings of the readers.
2. Because NOX is the main issue and involved in the title of this manuscript, more detailed information about structure, function, transcription/expression, regulation and localization of NOX isoforms are indispensable in the third section (3. Role of NOXs in Micturition).
3. As described in the 2nd section (2. Role of oxidative stress in DBD) oxidative stress seem to increase RNS rather than ROS under DM conditions. There are other sources for ROS production as described in the sentence at line121-123. The authors should mention why the NOX is shed light on in this article more clearly.
The authors should add figures and/or tables to summarize the contents in each section. In the present form, it is hard to understand the cause-and-effect relationships at once.
line 72-74; In the sentence “Higher levels of oxidative markers … were increased … “, “higher” seems redundant.
line 73; “MDA” should be spelled out fully.
line 94; does ‘deletion’ mean gene disruption? Please clarify the methodology.
line 122; ‘adenine’ is missing between ‘nicotinamide’ and ‘dinucleotide’.
line 134; “tissue-specific expression” seems better.
line 143; “enzymes” is redundant.
line 145; here ‘NOXs’ is better than ‘this family’.
lines 150 and 167; refine the style of ‘O2-‘‘.
line 189; ‘AGE’ should be spelled out fully.
line 195; PDBu should be spelled out fully.
line 212; ‘nox4’ should be in italics.
line 219; ‘wk’ should be spelled out fully.
Author Response
Firstly, we would like to thank the Reviewers for their time, consideration, critical comments, and helpful suggestions toward strengthening this review. We agree with most of their comments and have modified the text according to their main suggestions. We also thank the Editor for allowing us to submit a revised copy of the review.
Reply to Reviewer #2 comments:
The review article entitled “Targeting NADPH Oxidase as an Approach for Diabetic Blader Dysfunction” describes the recent progress in the research area of diabetic bladder dysfunction with a focus on its relationship with ROS and NOX. Briefly, the information provided is too specific and too rough to understand an overview of the role and significance of ROS and NOXs in diabetic blader dysfunction for the reader unfamiliar with this research area. In the present form this manuscript is not informative as a review article. The English should be rechecked carefully.
Thank you for your thoughtful and constructive feedback on our manuscript. We have carefully considered and we address your concerns bellow. English revision was performed by a native speaker. We hope you find this version suitable for publication.
Major issues
1.As mentioned in the first sentence of the 2nd section (2. Role of oxidative stress in DBD), generation of ROS/RNS depends on the balance of oxidants and antioxidants. In the present form information about antioxidants including regulation of NADP+/H level is missing. As a review, providing more well known general information will be helpful for the readers, such as comparisons of the results of other well characterized tissue/organ with those of bladder under DM conditions. Representation of the relationships of the factors inducing oxidative stress and phenomena with figures and/or tables will help the understandings of the readers.
First, as requested, we have included a brief description of antioxidants systems on page 2, line 62, where is now written:
“Most cells possess intrinsic defense mechanisms, involving various enzymes, such as superoxide dismutase (SOD), glutathione peroxidase (GPX) and catalase (CAT), as well as several reducing cofactors including NADH, NADPH, glutathione and tetrahydrofolate, whose levels help to manage redox potential [10,11].”
We have also included the following references for this subject:
-
- Koju, N., Qin, Zh. & Sheng, R. Reduced nicotinamide adenine dinucleotide phosphate in redox balance and diseases: a friend or foe?. Acta Pharmacol Sin43, 1889–1904 (2022). https://doi.org/10.1038/s41401-021-00838-7
- Chandel NS. NADPH-The Forgotten Reducing Equivalent. Cold Spring Harb Perspect Biol. 2021 Jun 1;13(6):a040550. doi: 10.1101/cshperspect.a040550.
Second, as requested, we have included new paragraph about the role of oxidative stress in the kidneys under DM conditions. Please see page 3, line 80, where is it written:
“For instance, in rodent models of diabetic kidney disease, oxidative stress has been associated with increased levels of nuclear factor kappa B (NF-kB), tumor necrosis factor alpha (TNF-α), and transforming growth factor beta-1 (TGF-β1) [21-24], while antioxidant systems are impaired [25-27]. This imbalance ultimately leads to renal glomerular sclerosis and interstitial fibrosis [27,28]. In isolated glomerular endothelial cells exposed to high glucose and serum from diabetic mice, there was increased mitochondria-derived ROS, secretion of pro-apoptotic factors, and reduced autophagy [29]. Clinical studies with DM patients further demonstrate that renal oxidative stress contributes to the progressive loss of the glomerular filtration barrier and fibrosis in the tubulointerstitial regions of the kidneys [30,31].”
We have also included the following references for this subject:
-
- Kapucu A. Crocin ameliorates oxidative stress and suppresses renal damage in streptozotocin induced diabetic male rats. Biotech Histochem. 2021 Feb;96(2):153-160. doi: 10.1080/10520295.2020.1808702.
- Kausar MA, Parveen K, Siddiqui WA, Anwar S, Zahra A, Ali A, Badraoui R, Jamal A, Akhter N, Bhardwaj N, Saeed M. Nephroprotective effects of polyherbal extract via attenuation of the severity of kidney dysfunction and oxidative damage in the diabetic experimental model. Cell Mol Biol (Noisy-le-grand). 2022 Jan 2;67(4):42-55. doi: 10.14715/cmb/2021.67.4.6.
- Biswas S, Chen S, Liang G, Feng B, Cai L, Khan ZA, Chakrabarti S. Curcumin Analogs Reduce Stress and Inflammation Indices in Experimental Models of Diabetes. Front Endocrinol (Lausanne). 2019 Dec 18;10:887. doi: 10.3389/fendo.2019.00887.
- Antar SA, Abdo W, Taha RS, Farage AE, El-Moselhy LE, Amer ME, Abdel Monsef AS, Abdel Hamid AM, Kamel EM, Ahmeda AF, Mahmoud AM. Telmisartan attenuates diabetic nephropathy by mitigating oxidative stress and inflammation, and upregulating Nrf2/HO-1 signaling in diabetic rats. Life Sci. 2022 Feb 15;291:120260. doi: 10.1016/j.lfs.2021.120260.
- Yang S, Fei X, Lu Y, Xu B, Ma Y, Wan H. miRNA-214 suppresses oxidative stress in diabetic nephropathy via the ROS/Akt/mTOR signaling pathway and uncoupling protein 2. Exp Ther Med. 2019 May;17(5):3530-3538. doi: 10.3892/etm.2019.7359.
- Hussain Lodhi A, Ahmad FU, Furwa K, Madni A. Role of Oxidative Stress and Reduced Endogenous Hydrogen Sulfide in Diabetic Nephropathy. Drug Des Devel Ther. 2021 Mar 5;15:1031-1043. doi: 10.2147/DDDT.S291591.
- Anwar, S.; Kausar, M.A.; Parveen, K.; Siddiqui, W.A.; Zahra, A.; Ali, A.; Saeed, M. A vegetable oil blend administration mitigates the hyperglycemia-induced redox imbalance, renal histopathology, and function in diabetic nephropathy. J. King Saud Univ. Sci.2022, 34, 102018. doi: 10.1016/j.jksus.2022.102018.
- Chen HW, Yang MY, Hung TW, Chang YC, Wang CJ. Nelumbo nucifera leaves extract attenuate the pathological progression of diabetic nephropathy in high-fat diet-fed and streptozotocin-induced diabetic rats. J Food Drug Anal. 2019 Jul;27(3):736-748. doi: 10.1016/j.jfda.2018.12.009.
- Casalena, G.A., Yu, L., Gil, R. et al. The diabetic microenvironment causes mitochondrial oxidative stress in glomerular endothelial cells and pathological crosstalk with podocytes. Cell Commun Signal 18, 105 (2020). doi: 10.1186/s12964-020-00605-x
- Ricciardi CA, Gnudi L. Kidney disease in diabetes: From mechanisms to clinical presentation and treatment strategies. Metabolism. 2021 Nov;124:154890. doi: 10.1016/j.metabol.2021.154890.
- Wang N, Zhang C. Oxidative Stress: A Culprit in the Progression of Diabetic Kidney Disease. Antioxidants (Basel). 2024 Apr 12;13(4):455. doi: 10.3390/antiox13040455.
Finally, in response to the request for additional figures and/or tables, we have included a new figure illustrating the potential sources of ROS under diabetic conditions (Figure 1 on page 2) in Section 2. Additionally, please note that we have already provided a graphical abstract and a figure depicting NOX expression in the pontine micturition center and lower urinary tract structures (now Figure 2, in Section 3).
2.Because NOX is the main issue and involved in the title of this manuscript, more detailed information about structure, function, transcription/expression, regulation and localization of NOX isoforms are indispensable in the third section (3. Role of NOXs in Micturition).
We have rewritten this paragraph to include additional information on NOX background. Please, see page 4, from line 145, where is now written:
“NADPH oxidases are transmembrane enzyme complexes that catalyze the reduction of O2 to ROS, O2- or H2O2, coupled to the oxidation of NADPH. Seven enzyme isoforms in the NOX family [NOX1, NOX2 (aka gp91phox), NOX3, NOX4, NOX5, DUOX1, and DUOX2] have been identified and found to be expressed in specific cell types and tissues, underlying their specific role in health and disease [50]. In humans, all seven enzymes are expressed, while mice and rats express NOX1–4 and DUOX isoforms but lack NOX5 [50, 52]. The expression levels of NOXs are frequently regulated by growth factors and cytokines as well as by other stimuli such as hypoxia, low pH, and bacterial products [53]. Their structure varies between isoforms but, in general, consist of a conserved 6-helix transmembrane domain and an N-terminal dehydrogenase domain located intracellularly, which is the binding sites for NADPH substrate and the co-factor FAD (flavin adenine dinucleotide). The oxidase complex is also regulated by cytosolic regulatory proteins, including p47, p67, p40 and the small GTPases Rac1 or Rac2, and, in the case of NOX5, DUOX1, and DUOX2 is also regulated by a calcium-binding domain [52]. For the interested reader, the sub-cellular localization and activation mechanism of NOXs has been reviewed elsewhere [53], as well as their isoform specific roles across different tissues and conditions, influencing various pathophysiological responses [51]. NOX-derived ROS contribute to physiological functions and processes, such as the oxidative burst of the innate immune response, cell proliferation, differentiation, migration, ion channel conductance, vasodilation, hearing, insulin secretion, and insulin sensitivity (for review see 9). Nevertheless, the role of NOX-derived ROS even in normal bladder function remains poorly understood. This gap highlights the need for targeted research to elucidate the specific contributions of NOXs in the urinary system.”
3.As described in the 2ndsection (2. Role of oxidative stress in DBD) oxidative stress seem to increase RNS rather than ROS under DM conditions. There are other sources for ROS production as described in the sentence at line121-123. The authors should mention why the NOX is shed light on in this article more clearly.
Thank you for your comment. We appreciate the feedback but are unsure if we have fully understood your suggestion correctly. The studies cited in Section 2 (Role of Oxidative Stress in DBD) describe the involvement of various reactive species, including both ROS and RNS, in DBD, and they do not conclusively state that RNS has a greater impact than ROS. Indeed, we have added references about RNS, but feel that that the main focus of the review is on ROS and NOXs.
Additionally, we already described in section 3 (Role of NOXs in Micturition) that “Unlike other ROS sources, NADPH oxidases are the only enzyme family known to produce ROS as their primary and sole function [50], and all other ROS sources serve distinct primary functions, and ROS production typically occurs after damage or uncoupling”. To make this clearer, we added the following statement on page 3, line 144: “This unique catalytic activity of NOXs is the rationale for a review highlighting their role in DBD”.
4.The authors should add figures and/or tables to summarize the contents in each section. In the present form, it is hard to understand the cause-and-effect relationships at once.
Please, see our comment to question #1.
Minor points
Line 72-74; In the sentence “Higher levels of oxidative markers … were increased … “, “higher” seems redundant.
Corrected.
Line 73; “MDA” should be spelled out fully.
Corrected.
Line 94; does ‘deletion’ mean gene disruption? Please clarify the methodology.
We changed “deletion” to “knockout” for clarity.
Line 122; ‘adenine’ is missing between ‘nicotinamide’ and ‘dinucleotide’.
Corrected.
Line 134; “tissue-specific expression” seems better.
Corrected.
Line 143; “enzymes” is redundant.
Corrected.
Line 145; here ‘noxs’ is better than ‘this family’.
Corrected.
Lines 150 and 167; refine the style of ‘O2-‘‘.
Corrected.
Line 189; ‘AGE’ should be spelled out fully.
Corrected.
Line 195; pdbu should be spelled out fully.
Corrected.
Line 212; ‘nox4’ should be in italics.
Corrected.
Line 219; ‘wk’ should be spelled out fully.
Corrected.
Round 2
Reviewer 2 Report
A revised version of the manuscript entitled “Targeting NADPH Oxidase as an Approach for Diabetic Bladder Dysfunction“ revised correctly, but requires a revision as pointed out in "Detail comments".
Line 204: it is not clear for me what "the scientific community" indicates. It will be better to rewrote the phrase "It is recognised ... community".
Author Response
Line 204: it is not clear for me what "the scientific community" indicates. It will be better to rewrote the phrase "It is recognised ... community".
We rewrote it to "The absence of validated isoform-specific antibodies and inhibitors significantly hinders investigations into the direct involvement of NOXs in the development and progression of various diseases."